# Whole-Blood Gene Expression Profiles Correlate with Response to Immune Checkpoint Inhibitors in Patients with Metastatic Renal Cell Carcinoma

**DOI:** 10.3390/cancers14246207

**Published:** 2022-12-15

**Authors:** Yoshiyuki Nagumo, Shuya Kandori, Takahiro Kojima, Kazuki Hamada, Satoshi Nitta, Ichiro Chihara, Masanobu Shiga, Hiromitsu Negoro, Bryan J. Mathis, Hiroyuki Nishiyama

**Affiliations:** 1Department of Urology, Faculty of Medicine, University of Tsukuba, Ibaraki 305-8577, Japan; 2Department of Urology, Aichi Cancer Center Hospital, Nagoya, Aichi 464-8681, Japan; 3International Medical Center, University of Tsukuba Affiliated Hospital, Ibaraki 305-8576, Japan

**Keywords:** whole-blood, gene expression, renal cell carcinoma, immune checkpoint inhibitor, response

## Abstract

**Simple Summary:**

Some metastatic renal cell carcinoma (mRCC) patients do not respond to immune checkpoint inhibitor (ICI) therapy. However, since predicting who will be a poor responder is difficult, a non-invasive, high-resolution genomic biomarker to accurately predict clinical responses to ICIs is urgently required. We found a minimum set of 14 genes that change in response to treatment and this panel can be used to accurately classify responder patients. Our results suggest that the gene signatures identified from a whole-blood transcriptome approach are clinically useful biomarkers for predicting ICI responses in patients with mRCC.

**Abstract:**

In metastatic renal cell carcinoma (mRCC), the clinical response to immune checkpoint inhibitors (ICIs) is limited in a subset of patients and the need exists to identify non-invasive, blood-based, predictive biomarkers for responses. We performed RNA sequencing using whole-blood samples prospectively collected from 49 patients with mRCC prior to the administration of ipilimumab (IPI) and/or nivolumab (NIVO) to determine whether gene expression profiles were associated with responses. An analysis from 33 mRCC patients with complete responses (*n* = 5), partial responses (*n* = 14), and progressive disease (*n* = 14) showed 460 differentially expressed genes (DEGs) related to immune responses between the responder and non-responder groups with significant differences. A set of 14 genes generated from the initial 460 DEGs accurately classified responders (sensitivity 94.7% and specificity 50.0%) while consensus clustering defined clusters with significantly differing response rates (92.3% and 35.0%). These clustering results were replicated in a cohort featuring 16 additional SD patients (49 total patients): response rates were 95.8% and 48.0%. Collectively, whole-blood gene expression profiles derived from mRCC patients treated with ICIs clearly differed by response and hierarchical clustering using immune response DEGs accurately classified responder patients. These results suggest that such screening may serve as a predictor for ICI responses in mRCC patients.

## 1. Introduction

The treatment of metastatic renal cell carcinoma (mRCC) has shifted rapidly with the sequential development of immune- and molecular-targeted therapies [1,2]. Notably, the combination of antibody-derived nivolumab ((NIVO), targeting anti-programmed death-1 (PD-1)), and ipilimumab ((IPI), targeting anti-cytotoxic T-lymphocyte-associated antigen 4 (CTLA-4)), has been established for mRCC patients with intermediate or poor risk as classified by the International Metastatic Renal Cell Carcinoma Database Consortium (IMDC) [3]. In contrast, NIVO is still used as a monotherapy for patients with progressive disease (PD) even after initial treatments with antiangiogenic agents [4]. Despite remarkable recent increases in mRCC survival, the proportion of PD patients after combination therapy (IPI + NIVO) and NIVO monotherapy is still high compared to patients treated with antiangiogenic agents: 20% vs. 17% and 35% vs. 26%, respectively [4,5]. As PD-1 and CTLA-4 blocking increases the number and efficacy of anti-tumor cytotoxic T cells, the observed high proportion of PD may stem from immune polymorphisms and other genetic factors that temper the effectiveness of such therapies [6]. Thus, non-invasive, high-resolution genomic biomarkers to accurately predict clinical responses to these immune checkpoint inhibitors (ICIs) are urgently required.

Until recently, no tissue- or blood-based biomarkers have been approved for mRCC patients for daily clinical practice. In terms of tissue-based biomarkers, tumor PD-L1 expression is used as an established diagnostic marker in patients with non-small cell lung cancer (NSCLC) [7] in addition to breast, head, and neck cancer patients [8,9]. However, while tumor mutation burden (TMB) and microsatellite instability (MSI) are considered useful biomarkers for ICI [10,11], the invasiveness of tissue-based measurements prevents their frequent use. Useful blood-based biomarkers, in contrast, could be more frequently sampled and provide enhanced resolution to therapy outcomes.

Herein, we show that whole-blood gene expression in mRCC patients treated with IPI and/or NIVO demonstrates clear differences between responder and non-responder groups. Furthermore, hierarchical clustering using 460 differentially expressed genes (DEGs) between the two groups, in line with a minimum set of 14 genes from these DEGs, accurately classified the responder patients. Our results suggest that the gene signatures identified from a whole-blood transcriptome approach are clinically useful biomarkers for predicting ICI responses in patients with mRCC.

## 2. Materials and Methods

### 2.1. Patients and Sample Collection

For our prospective study, we collected whole blood samples from 49 mRCC patients using PAXgene Blood RNA tubes (PreAnalytix) before the administration of IPI and/or NIVO at the University of Tsukuba Hospital between December 2016 and October 2020. All patients provided informed, written consent for the present study and the protocol was approved by the University of Tsukuba Hospital Institutional Review Board (#H28-104). Clinical data were obtained from hospital charts. Response to ICIs was evaluated according to Response Evaluation Criteria in Solid Tumor (RECIST) ver1.1. [12].

For analysis of the maximum difference in whole-blood gene expressions between the responder and non-responder groups, we first used data from 33 patients with the following responses defined as per RECIST: complete response (CR) (*n* = 5), partial response (PR) (*n* = 14), and progressive disease (PD) (*n* = 14) as the discovery dataset. The data from 16 remaining patients with stable disease (SD) were briefly used as the validation dataset to evaluate the performance of a gene set to classify clusters according to ICI responses. In the present study, responders to ICI were defined as CR, PR, and SD.

### 2.2. RNA Isolation and RNA Sequencing (RNA-seq)

Total RNA from whole-blood samples was isolated using a PAXgene Blood RNA kit (PreAnalytix) according to the manufacturer’s instructions. The globin RNA was removed from total RNA using a GLOBINclear-Human Kit (Thermo Fisher SCIENTIFIC). For RNA sequencing, 500 ng total RNA were rRNA-depleted using an NEBNext rRNA Depletion Kit (New England Biolabs, Hitchin, UK) according to the manufacturer’s instructions. Library preparation was performed using an NEBNext Ultra-Directional RNA Library Prep Kit (New England Biolabs). Libraries were validated via Bioanalyzer (Agilent Technologies, Santa Clara, CA, USA) to determine size distribution and concentration. Validated libraries were then sequenced at Tsukuba i-Laboratory LLP (Tsukuba, Japan) on a NextSeq500 (Illumina, San Diego, CA, USA) with the paired-end 36-base read option.

### 2.3. RNA-seq Analysis

Sequencing reads were mapped on the hg19 reference genome, quantified, and then normalized via quantile method using CLC Genomics Workbench version 10.1.1 (Qiagen, Aarhus, Denmark). DEG analyses were performed using the Empirical Analysis of DGE tools in CLC Genomics Workbench. The statistical threshold for DEGs was defined as a false discovery rate (FDR) < 0.1 or raw *p* value < 0.05, and absolute fold-change > 2.0. Principal component analysis (PCA) plots, heatmaps, and volcano plots were generated using R (version 4.0.2; R Foundation, Vienna, Austria).

### 2.4. Cluster Analysis

Hierarchical clustering based on Pearson distance and complete linkage was performed using the ComplexHeatmap package in R [13]. To improve the robustness of unsupervised hierarchical clustering evidence, the ConsensusClusterPlus package in R was also used to find an optimal cluster [14]. To generate the minimum number of genes from the DEGs that could accurately define clusters according to ICI responses, we used the prediction analysis of microarrays (PAM) package in R [15].

### 2.5. Enrichment Analysis

We analyzed enrichment in the DEGs using Metascape with default settings [16]. The gene set variation analysis (GSVA) package in R was also used to calculate sample-wise enrichment scores in the DEGs [17].

### 2.6. Statistical Analysis

Data are presented as means ± SD. For categorical variables, comparisons between groups were performed by Fisher’s exact probability test or Cochran–Armitage trend test. For continuous variables, we compared variables between groups using the Mann–Whitney U test. All statistical comparisons were two-sided and *p*-values < 0.05 were considered significant. GraphPad Prism8 (GraphPad Software, San Diego, CA, USA) was used for the statistical analyses.

### 2.7. Data Availability

Gene expression matrix processed using CLC Genomics Workbench and clinical data in the present study are available in the Appendix A.

## 3. Results

### 3.1. Patient Characteristics of the Discovery Dataset

Table 1 summarizes the characteristics of patients (*n* = 33) stratified by ICI response. No significant differences in the histology, IMDC classification, or treatment were found between the two groups. In contrast, the median age of responder patients was significantly younger than non-responder patients (*p* = 0.03). Furthermore, the proportion of patients with poor KPS was significantly higher in the non-responder group than the responder group (*p* = 0.0063).

### 3.2. Whole-Blood Gene Expression Profiles in Patients with ICI-Treated mRCC Were Significantly Different between Responders and Non-Responders

Figure 1 details the analysis workflow of the present study. To investigate differences in whole-blood gene expression profiles between responders and non-responders, we first performed a DEG analysis. Between the two groups, there were 34 DEGs in 33 patients with significant differences (two-fold changes (FC) and false discovery rate (FDR) <0.1) (Appendix A). Next, to investigate whether those DEGs could correctly classify patients into responder and non-responder groups, we performed a hierarchical clustering. As shown in Appendix A, these 34 DEGs could not clearly define two large clusters with differing responses.

### 3.3. Hierarchical Clustering Using 460 DEGs Clearly Defines Two Large Clusters with Differing Responses

To generate a gene set that could stratify clusters according to ICI response, we attempted another DEG analysis under the assumption that clusters could not be classified according to response alone since the number of DEGs used for hierarchical clustering was small. Therefore, by using raw *p* values (<0.05) instead of FDR (<0.1), 460 DEGs were newly identified (Appendix A). A PCA plot of the 460 DEGs showed a trend of cluster separation between the responders and non-responders (Figure 2A). Of these 460 DEGs, the responder group had 114 upregulated genes and 346 downregulated genes compared to the non-responder group (Figure 2B). Surprisingly, hierarchical clustering using 460 DEGs clearly defined two large clusters with distinctly differing responses (Figure 2C).

Regarding pathway and process enrichment, downregulated genes in the responder group were significantly enriched, especially in terms related to neutrophil degranulation and inflammatory responses, while upregulated genes were enriched in the Gene Ontology biological process codes (GOBPs) related to adaptive immune responses (Figure 3A and Appendix A). GSVA showed a significant difference in the enrichment of GOBPs related to innate and adaptive immune responses (Figure 3B and Appendix A). As shown in Figure 3C, the responder group demonstrated significantly higher enrichment scores in the representative four GOBPs associated with innate and adaptive immune responses compared to the non-responder group. These results suggest that whole-blood gene expression, especially with regard to the immune system, correlates with ICI response and allows further classification of clusters according to response.

### 3.4. Whole-Blood Gene Expression Profiles Stratified by ICI Type Were Also Significantly Different between Responders and Non-Responders

Since patients were treated with two types of therapies, i.e., the combination of IPI + NIVO and NIVO monotherapy, we next investigated whether whole-blood gene expression profiles differed between each type. The proportions of patients in the IPI + NIVO group and the NIVO group were 57.6% (*n* = 19) and 42.4% (*n* = 14), respectively. As shown in Figure 4A, the proportion of responders in the IPI + NIVO group tended to be higher than the NIVO group (63.1% vs. 50.0%). After the DEG analysis between the responder and non-responder groups, we identified 518 and 425 DEGs with significant differences (2-FC and raw *p* value < 0.05) in the IPI + NIVO group and the NIVO group, respectively (Appendix A). Hierarchical clustering using these 518 and 425 DEGs clearly defined two large clusters with differing responses in the corresponding groups (Figure 4B).

As shown in Figure 4C, a Venn diagram revealed only a small gene overlap (*n* = 34) between the two groups whereas most other genes were exclusive (Appendix A). The 34 overlapping genes were enriched in terms related to immune response, such as neutrophil degranulation, myeloid leukocyte migration, and humoral immune response (Appendix A). Although there were multiple exclusive DEGs between the IPI + NIVO group and the NIVO group, as shown in Figure 4C, a Circos plot showed that multiple genes between the two groups shared the same enriched terms (Figure 4D). Taken together, these results suggest that whole-blood gene expression profiles were markedly different according to ICI response in spite of different treatment types.

### 3.5. A Set of 14 Genes Could Accurately Classify Responders Treated with ICIs

Finally, we examined whether we could generate a minimal gene set that could accurately classify responders and non-responders since 460 DEGs are impractical to use in daily clinical or research activities. As shown in Figure 5A, using the nearest shrunken centroid classification in the PAM package, a set of 14 genes that best defined each cluster characterizing the corresponding response was generated. With these 14 genes, Fisher’s exact probability test showed a significant correlation for both the actual and predicted responses (*p* = 0.0049). The sensitivity and specificity of the 14-gene signature were 94.7% and 50.0%, respectively.

Next, to improve the robustness of our clustering, we used consensus clustering that provides solid quantitative and visual stability of unsupervised hierarchical clustering evidence [14]. As shown in Appendix A, the cumulative distribution function (CDF) plots at cluster number k = 2 showed an approximate maximum distribution of the consensus index, indicating maximum clustering stability. Figure 5B shows the consensus matrices for clusters at k = 2 and indicates a clear separation between the two clusters. Therefore, we used k = 2 as the optimal number for the robust stability of clustering. The consensus clustering heatmap in Figure 5C shows two clusters with significantly different response rates (92.3% and 35.0%), suggesting that this 14-gene set accurately classifies responders.

We then examined whether these 14 genes could accurately classify responders even if SD patients (*n* = 16) were mixed in with CR, PR, and PD patients (*n* = 33). As shown in Figure 6A, the consensus matrices for clusters at k = 3, which was the optimal number for the robust stability of clustering (Appendix A), indicated a clear separation between the two large clusters (total patients *n* = 49), while a consensus clustering heatmap showed two large clusters with significantly different response rates (95.8% and 48.0%) (Figure 6B). Taken together, this set of 14 genes accurately classifies responders even if the dataset is mixed with SD data.

## 4. Discussion

Here, we demonstrate that whole-blood-derived gene expression profiles from mRCC patients treated with ICIs show significantly differential responses and hierarchical clustering as evidenced by 460 DEGs between responder (CR and PR patients) and non-responder (PD patients) groups. Furthermore, even if SD patients are included, a set of 14 genes generated from the initial 460 DEGs can accurately define two large clusters with significantly differing response rates (95.8% and 48.0%).

For the evaluation of response to ICIs, tissue-based, functional biomarkers (such as TMB and MSI) are useful but invasive and can only provide temporal snapshots of response. Genomic screens, on the other hand, could lead to accurate predictions of response and side effects even before therapy begins if sufficiently broad and numerous datasets are collected for profiling. Until recently, there have been no such established blood-based biomarkers, including transcriptomics, that reliably predict ICI response in cancer patients [18]. In contrast, the fields of infectious and autoimmune diseases have repeatedly demonstrated that transcriptional blood profiling is useful for monitoring immune responses to pathogens, drugs, or vaccines [19,20,21]. For cancer patients, Friedlander et al. first reported that whole-blood transcriptomics could predict clinical response to tremelimumab (anti-CTLA-4) in patients with advanced melanoma using the largest recent cohort [22]. Regarding the predictive response, their novel classifier model using 15 genes had a favorable area under the curve of 0.86 (95% confidence interval. 0.81–0.91, *p* < 0.0001) in the independent validation cohort (*n* = 150). In line with that study, the present study shows that whole-blood transcriptomics correlates with ICI response in cancer patients to identify relevant gene signatures in spite of large differences in cancer type, drugs, and methodology.

From the initial 460 DEGs, we identified 14 key genes for classifying patients according to response: *RETN*, *MUC20*, *RP11-512M8.5*, *DEFA1B*, *CDKN3*, *ADAMTS2*, *ELOVL2*, *ZSCAN10*, *IL9R*, *MKRN3*, *GLTPD2*, *VIT*, *MYO1A*, and *EFCAB2* (Figure 5A). Importantly, PD-1 and CTLA4 were not significant in these results. However, to the best of our knowledge, few of these genes of interest have been studied in detail with regard to expression levels and immune responses. Of these 14 genes, 2 genes (*RETN* and *IL9R*) were included in the constituent genes for immune-related terms, such as neutrophil degranulation, innate immune response, and adaptive immune response. *RETN*, upregulated in the cluster enriched in non-responder patients (Figure 5C), codes for resistin, a secretory protein mainly produced by professional phagocytes [23], and is reported to be an inflammatory marker leading to the stimulation of the pro-inflammatory cytokines TNF-𝛼 and IL-12 via the NF-kB pathway [24]. Karapanagiotou et al. showed that serum resistin levels were higher in lung cancer patients compared with healthy controls [25] while Bonaventura et al. showed that higher serum resistin levels were associated with worse overall survival and that combining resistin levels and neutrophil-lymphocyte ratios could predict NIVO response in advanced NSCLC [26]. In terms of *IL9R* upregulation in the cluster enriched in responder patients, Wang et al. reported that Th9 cells, whose signature cytokine is IL-9, promote the proliferation of CD8^+^ T cells in an IL9R-dependent manner in colorectal cancer [27]. Therefore, higher levels of *IL9R* might contribute to anti-tumor effects through cytotoxic effector T cells and a shifting towards a favorable ICI response. Conversely, T cell populations that infiltrate tumors but fail to exert anti-tumor effects are referred to as dysfunctional and their presence may at least partially mediate ICI response [28]. Li et al. reported that, in human melanoma, CD8+ T cells continuously transition from effector T into dysfunctional T cell populations that express proliferation-associated genes during the early transition stages, including *CDKN3A* [29]. Such upregulated expression of *CDKN3A* in the non-responder group might promote poor ICI responses through the proliferation of these dysfunctional T cells in the tumor microenvironment. These reports suggest that key genes, such as *CDKN3A* and not all 14 potential genes, are associated with ICI responses through the regulation of the tumor immune milieu. As the other genes may provide fine tuning to the results, future studies detailing their contribution, as well as the polymorphism burden in the key genes, could enhance the predictive power of genomic scans, especially with regard to mRCC.

Our present study has several limitations due to the availability of clinical samples. First, there were no available samples for the independent validation cohort. Second, due to the small sample size, we could not separately generate a minimum gene set that accurately classifies responders by individual therapy. Third, in spite of whole-blood analysis, many confounding factors that affect response (such as selection bias) could not be excluded. However, our data do establish the potential of whole-blood genomic profiling with discrete gene sets as a practical target for daily clinical use in mRCC patients.

## 5. Conclusions

The present study demonstrated that whole-blood gene profiling in mRCC patients treated with IPI and/or NIVO showed clear differences that were predictive of their responses. Furthermore, hierarchical clustering using DEGs involved in immune responses between the responders and non-responders accurately classified the responder patients even if stable disease data were mixed in. Our results suggest that whole-blood gene expression profiling is attractive for predicting appropriate ICI candidates in patients with mRCC.

## Figures and Tables

**Figure 1 cancers-14-06207-f001:**
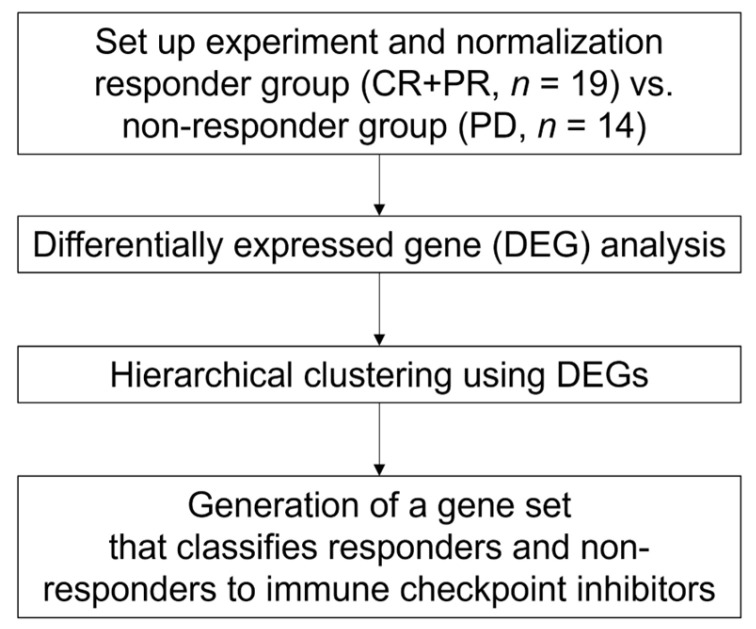
Analysis workflow of the present study.

**Figure 2 cancers-14-06207-f002:**
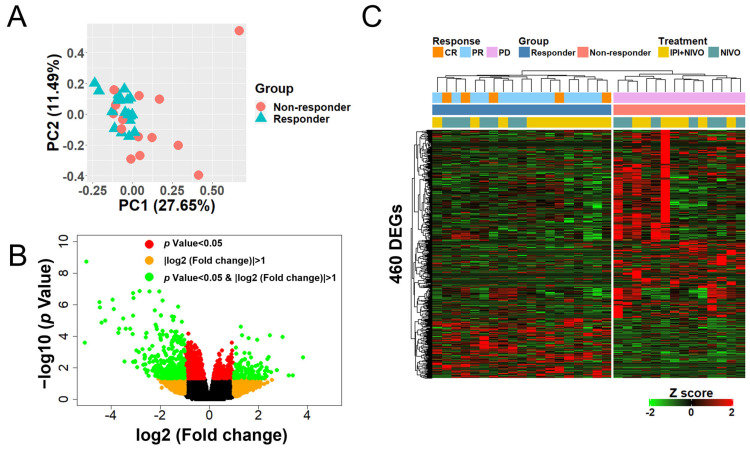
Hierarchical clustering using 460 differentially expressed genes (DEGs) clearly defines two large clusters with differing responses. (**A**) A principal component analysis plot showing the DEGs between the responder (blue) and non-responder groups (red). (**B**) A volcano plot showing the distribution of DEGs between the two groups is displayed in the indicated colors. (**C**) Unsupervised hierarchical clustering heatmap of 460 DEGs between the responder and non-responder groups.

**Figure 3 cancers-14-06207-f003:**
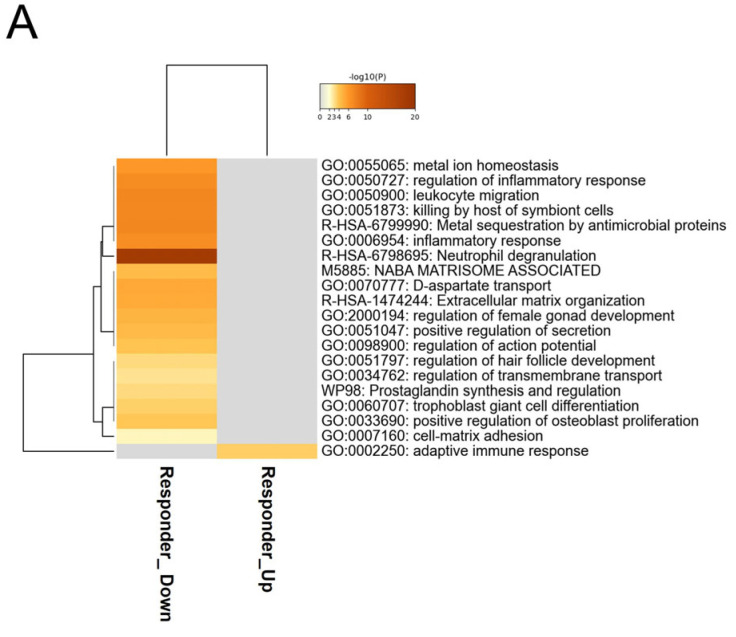
Whole-blood gene expression with regard to the immune system correlates with ICI response. (**A**) A heatmap of the significant Gene Ontology biological process (GOBP) categories for DEGs in each group. (**B**) A heatmap showing the distribution of sample-wise ESs in the 460 DEGs calculated using gene set variation analysis (GSVA). (**C**) Dot plots showing enrichment scores (ESs) calculated using GSVA between the two groups. Statistical analysis was performed by Mann–Whitney U test. (***, *p* < 0.001).

**Figure 4 cancers-14-06207-f004:**
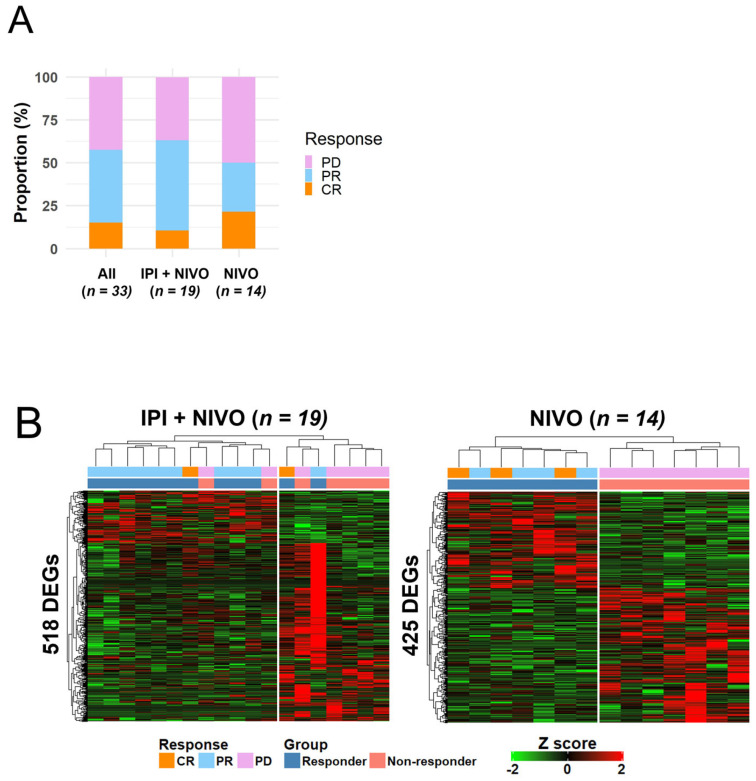
Whole-blood gene expression profiles stratified by ICI type were also different between responder and non-responder groups. (**A**) A stacked bar plot showing the proportion of patients with the indicated responses in the IPI + NIVO group and the NIVO group. (**B**) Hierarchical clustering heatmaps of DEGs between the responder and non-responder groups stratified by treatment type. (**C**) A Venn diagram showing the distribution of DEGs between the IPI + NIVO group and the NIVO group. (**D**) A Circos plot showing shared genes in the same ontology terms between the two groups (blue line).

**Figure 5 cancers-14-06207-f005:**
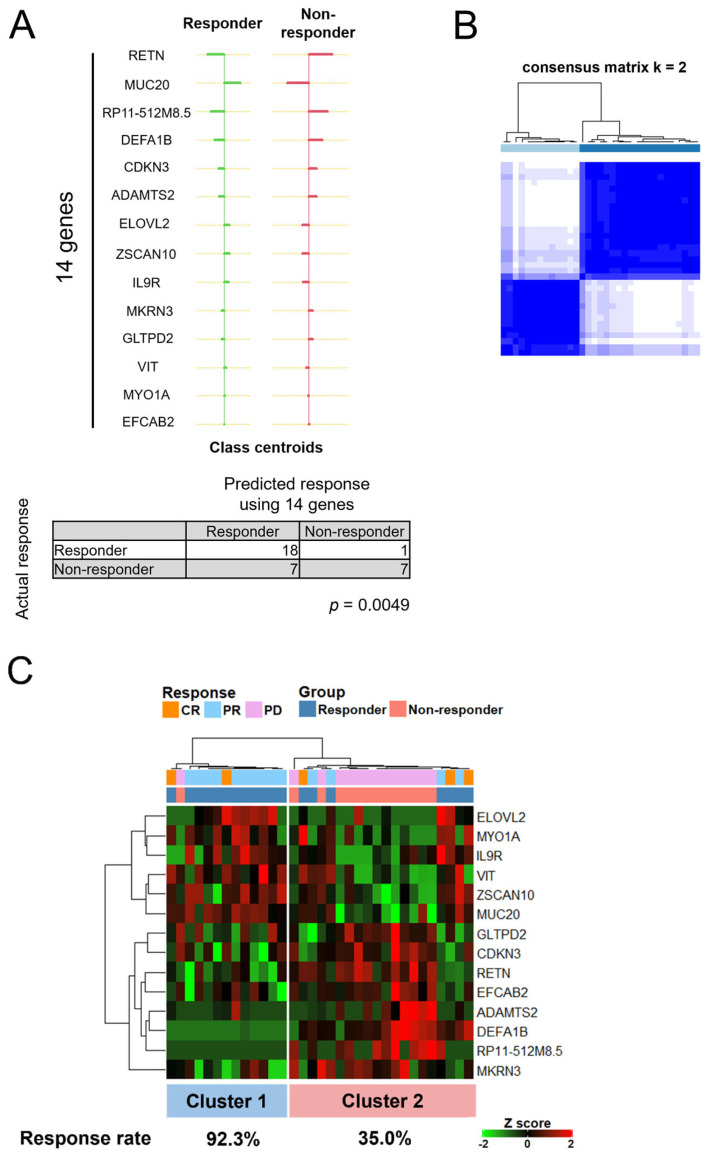
A set of 14 genes was generated that accurately classifies the clusters according to ICI response. (**A**) A set of 14 genes was generated by prediction analysis of microarrays (PAM) using the nearest shrunken centroid method. The class error rate of each group was calculated by PAM. Statistical analysis was performed by Fisher’s exact test. (**B**) Consensus matrices for clusters number at k = 2 was analyzed by ConsensusClusterPlus using 14 genes. (**C**) A consensus clustering heatmap of 14 genes in 33 patients with complete response (CR), partial response (PR), and progressive disease (PD).

**Figure 6 cancers-14-06207-f006:**
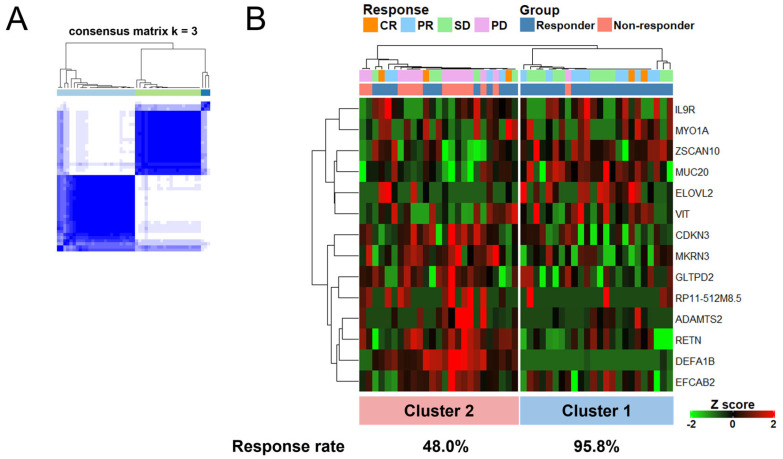
A set of 14 genes accurately classifies responders in the validation dataset mixed with SD data. (**A**) Consensus matrices for clusters number at k = 3 analyzed by ConsensusClusterPlus using 14 genes. (**B**) A consensus clustering heatmap of 14 genes in 49 patients with complete response (CR), partial response (PR), progressive disease (PD), and stable disease (SD).

**Table 1 cancers-14-06207-t001:** Patient characteristics of the discovery dataset (*n* = 33).

		Responders (CR/PR)	Non-Responders (PD)	*p* Value
Number of Patients		19	14	
Sex	Male	17 (89.5)	9 (64.3)	0.11
	Female	2 (10.5)	5 (35.7)	
Median Age (Range)		66.0 (46–82)	71.5 (61–84)	0.03
Histology	Clear Cell Carcinoma	15 (78.9)	11 (78.6)	0.70
	Clear Cell Carcinoma + Sarcomatoid Variant	2 (10.5)	1 (7.1)	
	Sarcomatoid	1 (5.3)	1 (7.1)	
	Papillary	1 (5.3)	0	
	Xp 11 Translocation	0	1 (7.1)	
IMDC Classification	Favorable	4 (21.1)	0	0.10
	Intermediate	11 (57.9)	9 (64.3)	
	Poor	4 (21.1)	5 (35.7)	
Treatment	Nivolumab	7 (36.8)	7 (50.0)	0.50
	Ipilimumab + Nivolumab	12 (63.2)	7 (50.0)	
KPS	100	14 (73.7)	4 (28.6)	0.0063
	90	2 (10.5)	4 (28.6)	
	80	3 (15.8)	1 (7.1)	
	70 > =	0	5 (35.7)	

CR: complete response, PR: partial response, PD: progressive disease, IMDC: International Metastatic Renal Cell Carcinoma Database Consortium, KPS: Karnofsky Performance Status.

## Data Availability

The datasets presented in this study can be found in online repositories. The names of the repository/repositories can be found in the article/supplementary material.

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
