# Peer review of "Whole-Blood Gene Expression Profiles Correlate with Response to Immune Checkpoint Inhibitors in Patients with Metastatic Renal Cell Carcinoma"

_cancers, 2022, doi:10.3390/cancers14246207_

Round 1

Reviewer 1 Report

This article presents a study to look at differential gene expression as a function of ICI.  The methodology and results are appropriate for this type of study.  My recommendation is for the authors NOT to consider whole-blood as a biomarker.  The 14 genes that they identified are consider the biomarkers.  "definition: biomarker is a measurable substance in an organism whose presence is indicative of some phenomenon such as disease, infection, or environmental exposure."  This should be corrected in the manuscript.

Reviewer 2 Report

The authors used pre-treatment blood transcriptome to predict response to immune checkpoint inhibitor therapy in metastatic renal cell carcinoma patients. In the discovery sample, 33 patients were involved. Another 16 patients were used to validate the 14 selected gene biomarker panel.

The study design is fine but the presentation and statistics could be improved and make the results more clearly understandable.

1. Discovery and validation patients datasets

page 2. Line 78-83. The author could better say that the 33 patients formed the discovery dataset and the 16 patients formed the validation dataset. 

 "Another 16 patients with stable disease (SD) formed a validation dataset to evaluate the performance of the 14 genes panel biomarker ...."

"3.1. Patient characteristics of the discovery dataset"

2. Figure 2C. How the clustering was done ? by supervised or unsupervised method ?

3. What are the 34 genes (overlapping in the Venn diagram) in Figure 3C ?

4. Response rate was used in Figure 4. It could be confusing. Why not use the standard test performance parameters, sensitivity, specificity, accuracy and F1 score and ROC-AUC etc. to report the ability to differentiate Responder vs non-responders.

5. Figures 3 and 4 have too many component parts. They could be separated into some more individual figures. (instead of being A, B, C, D, E etc of figure 3).

A check in GEO showed that 

Accession "GSE205003" is currently private and is scheduled to be released on May 31, 2024.
